# Hypochloremia, non-medication related associated factors, and impact on clinical outcomes in patients with acute heart failure: Insights from resource limited setups

Belay Mengistu Gebrie[1], Molla Asnake Kebede[2]*, Kedir Negesso Tukeni[2], Mohammed Mecha Abafogi[2], Turi Abateka Abadiga[3], Bethelhem Yaynemsa Sequr[1], Biniyam Beyene Tabor[4], Selemon Gebrezgabiher Asgedom[5]

1 Department of Internal Medicine, School of Medicine, College of Health Sciences and Medicine, Mizan - Tepi University, Mizan Teferi, Ethiopia, 2 Department of Cardiology, School of Medicine, College of Health Sciences and Medicine, Jimma University, Jimma, Ethiopia, 3 Department of Pharmacy, Mizan Tepi university Teaching Hospital, Mizan Aman, Ethiopia, 4 Department of Radiology, St. Paul's Hospital Millennium Medical College, Addis Ababa, Ethiopia, 5 School of Medicine, Yekatit 12 Hospital Medical College, Addis Ababa, Ethiopia

* mollaasnake75@gmail.com

## Abstract

### Background

Electrolyte disturbances such as hypochloremia are common in patients with acute heart failure (AHF) and may worsen clinical outcomes. However, data from low-resource settings remain limited. Therefore, the aim of this study is to determine the prevalence of hypochloremia among AHF patients, assess its association with length of hospital stay and in-hospital mortality, and identify factors independently associated with its occurrence.

### Objective

To determine the prevalence of hypochloremia among patients admitted with acute heart failure, to assess its non-medication association with length of hospital stay and in-hospital mortality, and to identify factors independently associated with hypochloremia in a resource-limited setting.

### Methods

A retrospective observational cohort study of hospitalized patients with acute heart failure, with serum chloride measured at admission was conducted among 260 patients aged ≥16. Data were analysed using SPSS version 26.0. The association between hypochloremia and clinical outcomes, including in-hospital mortality and length of hospital stay, was assessed using the Chi-square test and the Mann–Whitney U test, respectively. Multivariable logistic regression analysis was performed to

**Data availability statement:** Data cannot be shared publicly due to the presence of patient-related information, but the data are available from the Jimma University Research Data Repository upon a reasonable request, made by contacting the health institute's official email at ero@ju.edu.etc.

**Funding:** This research was funded by Jimma University solely to support the conduct of the study. No specific grants were received for authorship or publication from any public, commercial, or not-for-profit funding agencies. The funders had no role in study design, data collection and analysis, decision to publish, or preparation of the manuscript Reference: JMU 222/25.

**Competing interests:** The authors have declared that no competing interests exist.

identify independent predictors of hypochloremia. A p-value of <0.05 was considered statistically significant.

## Results

The prevalence of hypochloremia was 33.1% (95% CI: 27.4%–39.2%), and hypochloremic patients had significantly longer hospital stays (median: 12 days vs. 8.5 days; p = 0.001) and higher in-hospital mortality ($\chi^2$ = 8.58; p = 0.003). Multivariate analysis showed that NYHA class IV heart failure [AOR = 6.96; 95% CI: 1.49–32.4; p = 0.014], history of COPD [AOR = 4.94; 95% CI: 1.36–17.9; p = 0.001], hyponatremia [AOR = 2.20; 95% CI: 1.8–9.5; p = 0.001], and hypokalemia [AOR = 4.08; 95% CI: 1.53–10.6; p = 0.004] were significantly associated with hypochloremia.

## Conclusion

In this study hypochloremia at admission was common and associated with higher in-hospital mortality and longer hospital stays. It was more frequently observed in patients with severe heart failure, COPD, hypokalemia and hyponatremia.

## Background

Acute heart failure (AHF) is a complex clinical syndrome characterized by rapid onset or worsening of heart failure symptoms, often leading to emergency hospitalization. It is a major public health concern with high morbidity, mortality, and resource utilization globally, particularly in low- and middle-income countries [1,2]. In-hospital mortality from AHF remains significant despite therapeutic advancements, and identifying early prognostic markers is crucial for improving outcomes [3].

Electrolyte imbalances are frequent among patients with AHF, influenced by neurohormonal activation, renal dysfunction, and pharmacologic interventions such as loop diuretics. While disturbances in sodium and potassium are commonly evaluated, the prognostic value of chloride has received comparatively little attention. Emerging evidence suggests that hypochloremia, defined as serum chloride levels below 98 mmol/L, may serve as an independent predictor of poor outcomes in both chronic and acute heart failure settings [3,4]. Several mechanisms may explain this association. Hypochloremia is known to activate the renin–angiotensin–aldosterone system (RAAS), reduce responsiveness to diuretics, and reflect severe neurohormonal derangement factors contributing to worse prognosis [5,6]. Studies have linked hypochloremia with increased in-hospital mortality, longer length of hospital stay, higher rates of diuretic resistance, and readmissions [7,8].

Despite these findings, there is a paucity of data regarding the prevalence, predictors, and clinical impact of hypochloremia among patients with AHF in sub-Saharan Africa. This gap is particularly relevant in Ethiopia, where diagnostic and treatment capacities are variable, and disease patterns may differ from those in high-income settings. Therefore, this study aimed to assess the association between

hypochloremia and in-hospital mortality and length of hospital stay in AHF patients, and to identify independent predictors of hypochloremia in this population.

## Method and materials

### Study setting and period

The study was conducted at Jimma Medical Center (JMC), a major teaching and referral hospital located in Jimma Town, Oromia Region, Southwest Ethiopia. JMC serves an estimated population of approximately 2.77 million people from the surrounding zones. Data were collected from March 27 to May 30, 2025, and included patient charts for individuals admitted between December 1, 2022, and December 1, 2024.

### Study design and population

A retrospective observational cohort study of hospitalized patients with acute heart failure, with serum chloride measured at admission. It was conducted among patients aged ≥16 years to JMC during the study period.

**Eligibility criteria.** All ≥ 16 years old, patients with acute heart failure who admitted to Jimma university medical center and determined serum chlorine level on admission. Patients without serum chloride levels at admission were excluded.

### Sample size determination and sampling technique

A single population proportion formula was used to calculate the sample size by using 36.7% of prevalence of hypochloremia among acute heart failure patients in Ethiopia, 95% confidence interval (CI), and margin of error of 5%.

$$n = \frac{(Z\,\alpha/2)^2 * p(1-p)}{d^2}$$

Where
n = sample size
Z = cut off value of the normal distribution at 95% CI = 1.96
P = proportion of hypochloremia among acute heart failure patients = 0.367
d = marginal error = 0.05

$$n = \frac{(1.96)^2 * 0.367(1-0.367)}{(0.05)^2}$$

n = 357

The source population was less than 10,000(N = 700 from internal hospital statistics in the study period). Then sample size was corrected by using a correction formula.

$$\text{Corrected sample size} = \frac{n}{1 + n/N} = 260$$

Systematic random sampling technique was used. k is calculated as N/n and every 2 interval, randomly selected patients who fulfill inclusion criteria were included until the calculated sample size is obtained (Fig 1)

For our secondary objective, factors associated with hypochloremia were estimated using the double population proportion formula in Epi Info Version 7 (STATCALC), considering variables such as hyponatremia, NYHA class IV, and loop diuretic use from previous studies [9,10,11]. The largest calculated sample (n = 126) was smaller than the primary

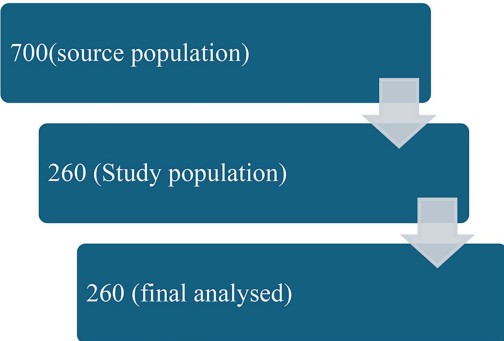

**Fig 1. Flow diagram of study population selection.** Flowchart illustrating the screening, eligibility assessment, exclusions, and final inclusion of patients admitted with acute heart failure at Jimma Medical Center between December 2022 and December 2024. A total of 260 patients were evaluated for eligibility. The diagram details the number of patients excluded and the final number included in the analysis.

objective sample; therefore, the final sample size remained 260. A systematic random sampling technique was applied to select study participants.

## Study variables

Data was collected on a range of socio-demographic, clinical, laboratory, treatment, and outcome variables. Socio-demographic variables included age, sex, and residence. Clinical variables comprised type of heart failure (acute decompensated or new onset), NYHA functional class, underlying etiology, left ventricular ejection fraction (LVEF), vital signs, pulmonary congestion, peripheral edema, and comorbidities such as hypertension, diabetes mellitus, chronic obstructive pulmonary disease (COPD), atrial fibrillation, and anemia.

Laboratory variables included serum chloride, sodium, potassium, hemoglobin, creatinine, and blood urea nitrogen (BUN). The primary variable, serum chloride, was measured at admission as a continuous variable, with hypochloremia defined as serum chloride <98 mmol/L. Treatment variables covered the use of loop diuretics, renin-angiotensin-aldosterone system (RAAS) inhibitors, beta-blockers, SGLT2 inhibitors, statins, and aspirin.

**Primary exposure.** Hypochloremia at admission

**Primary outcomes.** In-hospital mortality

length of hospital stays

**Operational definition.** Acute Heart Failure (AHF): Rapid onset or worsening of heart failure symptoms requiring hospitalization, including both new onset and acute decompensated heart failure cases [12].

Hypochloremia: Serum chloride level less than 98 mmol/L measured at admission [13].

NYHA Functional Class: Classification of heart failure severity based on symptoms during physical activity, categorized as Class I–IV [12].

Heart Failure Types: [12]

HFrEF: LVEF <40%

HFmrEF: LVEF 41–49%

HFpEF: LVEF ≥50%.

Length of Hospital Stay: Number of days from admission to discharge or in-hospital death [14].

In-hospital Mortality: Death occurring during hospital admission [15].

Pulmonary Congestion: Clinical or radiological evidence of fluid accumulation in lungs [12].

Peripheral Edema: Swelling in lower extremities detected on physical exam [16].

Hyponatremia: Serum sodium level less than 135 mmol/L [17].

Hypokalemia: Serum potassium level less than 3.5 mmol/L [18].

## Data collection procedure

Data were collected retrospectively using a structured English checklist that captured patient demographics, clinical characteristics, laboratory results, comorbidities, treatments, and outcomes. Two trained internal medicine residents extracted data from medical records using the Kobo Toolbox application under close supervision of the principal investigator to ensure data accuracy and completeness.

## Data quality control

Data quality was ensured throughout the study. The checklist was pretested on 5% of the sample at a separate hospital to ensure clarity and applicability. Data collectors received comprehensive training before data extraction. Completed checklists were reviewed daily for completeness, consistency, and accuracy, and discrepancies were corrected promptly. Prior to analysis, data were cleaned and cross-checked for errors. The principal investigator oversaw data management and secure storage throughout the study period. All included patients (n = 260) had complete data for serum chloride and all covariates included in the analysis. No missing data was observed for the variables analyzed, as a result no imputation methods were applied.

## Data analysis

Data were exported to SPSS version 26.0 for analysis. Descriptive statistics were used to summarize the data: means and standard deviations for continuous variables, and frequencies with percentages for categorical variables. Differences in in-hospital mortality between hypochloremic and non-hypochloremic patients were analyzed using the Chi-square test. The Shapiro-Wilk test assessed data normality, and since the length of hospital stay was non-normally distributed, comparisons were made using the Mann-Whitney U test. Bivariate logistic regression was used to explore associations between independent variables and hypochloremia. Variables with a p-value < 0.25 were included in the multivariable binary logistic regression model to determine independent predictors. Results were reported as adjusted odds ratios (AORs) with 95% confidence intervals (CIs), and statistical significance was set at $p < 0.05$. Model fitness was evaluated using the Hosmer-Lemeshow goodness-of-fit test.

## Ethical statement

Ethical clearance was obtained from the Institutional Review Board of Jimma University Institute of Health. Ref. number: JUIH/ IRB/ 222/25. This study was conducted in accordance with the principles of the Declaration of Helsinki. The study used retrospective data collected from medical records; therefore, the Institutional Review Board of Jimma University waived the requirement for informed consent from individual patients. No personal identifiers were recorded, and all data were fully anonymized prior to analysis. Collected data were securely stored in the Jimma University research data repository and can be accessed upon request via the office email at ero@ju.edu.et and strict confidentiality was maintained throughout the study. The anonymized data set is attached to this manuscript.

## Results

### Socio-demographic characteristics

A of total 260 acute heart failure patients who had a determination of chloride level on admission were included in this study. The age of Participants ranged from 16 to 90 years with mean age 51.6 ± 18.4SD. The majority of the study participants 156 (60%) were male and 104(40%) were female. Out of total study subjects 184(70.8%) were rural residents and 76(29.2%) were urban residents.

## Clinical characteristics

Out of the total 260 study participants, 187(71.9%) had acute decompensated heart failure whereas 73(28.1%) had new onset (de novo) heart failure and almost all study participants 259(99.6%) had stage C heart failure. Most common current underlying HF diagnosis was ischemic heart disease 89(34.2%) followed by CRVHD 86(33.4%) and the least common underlying HF diagnosis was RCMP 4(1.5%). 118(45.8%) of study participants had a history of hospitalization due to heart failure. Majority of the study participants, 175(67.3%) had NYHA class III whereas 5 (1.9%) had NYHA class I. The mean LVEF of the study participant was 45.5±15SD with minimum 18 and maximum 70. Overall, 119(45.8%) had heart failure with reduced ejection fraction (HFrEF), 116(44.6%) had heart failure with preserved ejection fraction (HFpEF), and 25(9.6%) had heart failure with mildly reduced ejection fraction. Nearly half of study subjects, 134 (51.5%, had pulmonary congestion.

The mean SBP value and mean DBP value of study subjects were 111.3±17.1 mmHg SD and 69.7±10.2 mmHg SD respectively. Of the total study subjects, 61(23.5%) was greater than 120 mmHg systolic blood pressure, 7(2.7%) had less than 90 mmHg systolic blood pressure, 34(13.1%) had greater than 80 mmHg diastolic blood pressure, 13(5.0%) had less than 60 mmHg diastolic blood pressure, 93(35.8%) had pulse rate greater than 100 beat/minute. Majority of the participants, 244(93.8%) had peripheral edema (Table 1)

Regarding the comorbidities, out of total participants, 96(36.9%) had hypertension, 29(11.2%) had diabetic mellitus, 21(8.1%) had chronic kidney disease, 16(6.2%) had chronic obstructive pulmonary disease(COPD), 78(30.0%) had atrial fibrillation, 89(34.2%) had anemia, 10(3.8%) had thyrotoxicosis, and only one study participant had retroviral infection (Fig 2)

## Laboratory data

The mean value serum sodium level was 135.9 mmol/L with standard deviation of 5.9 mmol/L and 90(34.6%) of the participants had low serum sodium level. The mean level of serum potassium of study subjects was 4 mmol/L±0.7 mmol/L SD and abo

ut 43(16.5%) had hypokalemia. The minimum and maximum values of hemoglobin level in this group of patients were 5.6g/dl and 18 g/dl, respectively with mean of 12.5g/dl and standard deviation of 2.2g/dl. The mean levels of serum creatinine and blood urea nitrogen were 1.2 mg/dl±1.1 mg/dl SD and 38.4 mg/dl±26 mg/dl SD respectively.

## Treatment related characteristics

Out of the total, 252(96.9%) of study participants were on diuretics treatment. Out of these 252 participants, all of them were on loop diuretics and only 2 of study participants on thiazide diuretics. The mean daily dose of loop diuretics that study participants received was 92.3 mg±39.7 mg SD, with a minimum of 20 mg/day and a maximum of 240 mg/day. 167(64.2%) of study subject were treated with renin-angiotensin-aldosterone system inhibitors, 227(87.3%) were treated with beta blockers, 33(12.7%) were on sodium-glucose cotransporter-2 (SGLT2) inhibitors, 127(48.8%) were on statin (atorvastatin), 128(49.2%) were on aspirin.

The mean length of hospital stay was 11.2±6.9days SD with minimum, 2 days and maximum, 48 days. Out of total participants, 239 (91.9%) were discharged with improvement and 21(8.1%) in hospital deaths were observed.

## Prevalence of hypochloremia

The minimum and maximum values of serum chloride level were 70 mmol/L and 136 mmol/L, respectively with mean of 99.3 mmol/L and standard deviation of 8.4 mmol/L. Out of the total study participants, hypochloremia was observed in 86(33.1%) of study participants. Out of those who had hypochloremia at admission, hypochloremia of 45(52.3%) participants were corrected and hypochloremia of 41(47.7%) participants were not corrected after admission.

**Table 1. Clinical characteristics among adult patients with acute heart failure admitted to Jimma Medical Center, December 2022–December 2024 (n = 260).**

| Variables | Frequency (%) |
|---|---|
| Current underlying HF diagnosis | |
| ACS | 26(10%) |
| Corpulmonale | 5(1.9%) |
| CRVHD | 86(33.1%) |
| DCMP | 45(17.3%) |
| HHD | 5(1.9%) |
| IHD | 89(34.2%) |
| RCMP | 4(1.5%) |
| Category of Heart failure | |
| Acute decompensated heart failure | 187(71.9%) |
| De novo (new onset) | 73(28.1%) |
| NYHA functional classification | |
| Class I | 5(1.9%) |
| Class II | 43(16.5%) |
| Class III | 175(67.3%) |
| Class IV | 37(14.2%) |
| Heart failure type | |
| HFmrEF | 25(9.6%) |
| HFpEF | 116(44.6%) |
| HFrEF | 119(45.8%) |
| Systolic blood pressure category | |
| 90 to 120 mmHg | 192(73.8%) |
| Less than 90 mmHg | 7(2.7%) |
| Greater than 120 mmHg | 61(23.5%) |
| Diastolic blood pressure category | |
| 60 to 80 mmHg | 213(81.9%) |
| Less than 60 mmHg | 13(5.0%) |
| Greater than 80 mmhg | 34(13.1%) |
| Pulse rate category | |
| Less than 100 beat/min | 167(64.2%) |
| Greater than 100 beats/min | 93(35.8%) |

*NB:* HF = heart failure, ACS = acute coronary syndrome, CRVHD = chronic rheumatic valvular heart disease, DCMP = dilated cardiomyopathy, HHD = hypertensive heart disease, IHD = ischemic heart disease, RCMP = restrictive cardiomyopathy, NYHA = New York Heart Association, HFmrEF = heart failure with mildly reduced ejection fraction, HFpEF = heart failure with preserved ejection fraction, and HFrEF = heart failure with reduced ejection fraction.

## Outcomes of hypochloremia in patients with AHF

he results showed that in-hospital death was significantly higher among patients with hypochloremia (15.1%) compared to those with normal serum chloride levels (4.5%) ($\chi^2(1) = 8.58$, p = 0.003). The test confirmed that this difference was statistically significant. The Kolmogorov–Smirnov and Shapiro-Wilk tests were used to assess the normality of the length of hospital stay, and the data were found to be non-normally distributed even after log transformation. Therefore, a Mann-Whitney U test was employed. The analysis revealed that patients with hypochloremia had a significantly longer hospital stay (median: 12 days) compared to those with normal chloride levels (median: 8.5 days) (p = 0.001).

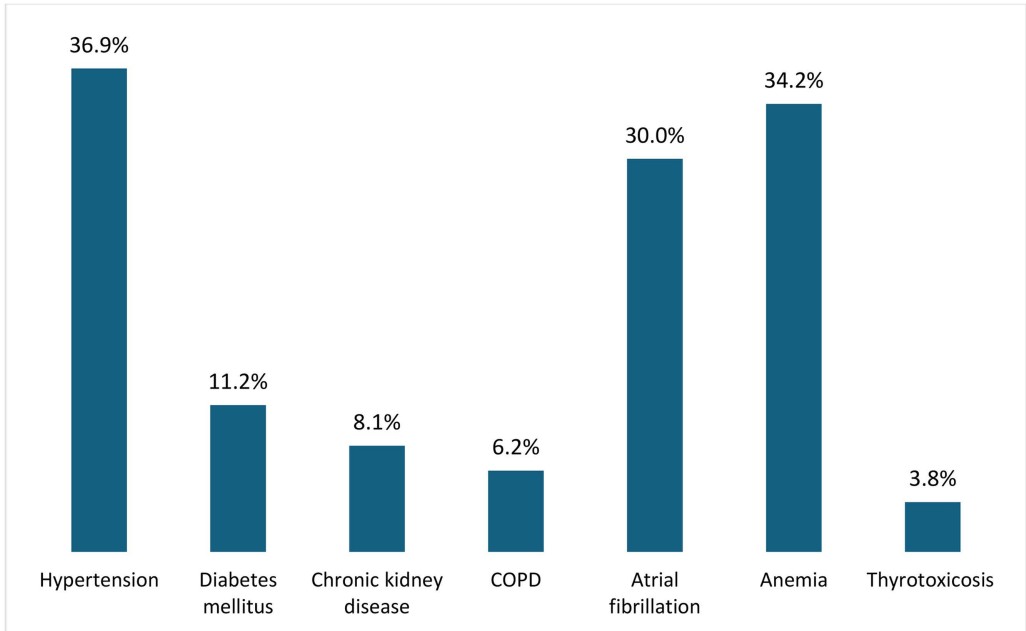

**Fig 2. Prevalence of comorbidities among adult patients admitted with acute heart failure.** Bar chart showing the distribution and prevalence of comorbid conditions among 260 adult patients admitted with acute heart failure at Jimma Medical Center between December 2022 and December 2024. COPD = Chronic Obstructive Pulmonary Disease.

### Factors associated with hypochloremia among AHF patients

The outcome of multivariate logistic regression stated that NYHA classification HF, history of COPD, hyponatremia, and hypokalemia were factors significantly associated with hypochloremia at p < 0.05. Patients with NYHA class IV heart failure were seven times more likely to develop hypochloremia when compared to NYHA class I and II heart failure patients [AOR = 6.96, 95%CI: 1.49–32.4, p 0.014]. The odds of developing hypochloremia among heart failure patients having history of chronic obstructive pulmonary disease were five times higher as compared to those who didn't have history of COPD [AOR 4.94, 95%CI: 1.36–17.9, p 0.001]. Compared to HF patients with normal sodium, those with hyponatremia were 2.2 times more likely to develop hypochloremia [AOR 2.2, 95%CI: 1.8–9.5, p 0.001]. Heart failure patients with hypokalemia have four times odds of developing hypochloremia as compared to those normal potassium [AOR 4.08, 95%CI: 1.53–10.6, p 0.004] (Table 2).

### Discussion

Despite the significant clinical impact of hypochloremia among heart failure (HF) patients, there remains limited data on its prevalence and associated factors among acute heart failure (AHF) patients in Africa. This study aimed to fill that gap and revealed alarming findings: one-third of AHF patients had hypochloremia. Furthermore, hypochloremia was associated with a longer length of hospital stay and higher in-hospital mortality. Age, NYHA functional class IV, history of chronic obstructive pulmonary disease (COPD), hyponatremia, and hypokalaemia were found to be significantly associated with hypochloremia in this population. The global burden of heart failure and related electrolyte disturbances highlights the clinical importance of these findings [19,20].

The prevalence of hypochloremia among acute heart failure (AHF) patients in the current study was 33.1% (95% CI: 27.4%–39.2%), which is considerably higher than reported in the United Kingdom (10.7%), Japan (12.5%) [21], the United

**Table 2. Factors associated to hypochloremia among adult patients with acute heart failure admitted to Jimma Medical Center, December 2022–December 2024 (n = 260).**

| Variables | Categories | Hypochloremia | | COR(95%CI) | P value | AOR(95%CI) | p-value |
|---|---|---|---|---|---|---|---|
| | | Yes | No | | | | |
| NYHA classification | Class I & II | 5 | 43 | 1 | | 1 | |
| | Class III | 61 | 114 | 4.6(1.73-12.3) | 0.453 | 3.05(0.85-10.9) | 0.087 |
| | Class IV | 20 | 17 | 10.1(3.72-31.3) | 0.089 | 6.96(1.49-32.4) | **0.014*** |
| Hospitalization history due to HF | No | 39 | 103 | 1 | | 1 | |
| | Yes | 47 | 71 | 1.74(1.03-2.9) | 0.341 | 1.75(0.82-3.73) | 0.15 |
| Type of HF | HFpEF | 30 | 86 | 1 | | 1 | |
| | HFrEF | 56 | 88 | 1.82(1.07-3.11) | 0.103 | 1.29(0.58-2.87) | 0.53 |
| Pulmonary congestion | No | 30 | 96 | 1 | | 1 | |
| | Yes | 56 | 78 | 2.29(1.35-3.92) | 0.212 | 1.83(0.86-3.9) | 0.12 |
| Systolic blood pressure | ≤ 120 mmHg | 71 | 128 | 1 | | 1 | |
| | > 120 mmHg | 15 | 46 | 0.59(0.31-1.13) | 0.153 | 0.59(0.22-1.64) | 0.32 |
| Diastolic blood pressure | ≤ 80 mmHg | 80 | 146 | 1 | | 1 | |
| | > 80 mmHg | 6 | 28 | 0.39(0.16-0.98) | 0.095 | 0.36(0.09-1.44) | 0.15 |
| Pulse rate | ≤ 100/min | 48 | 119 | 1 | | 1 | |
| | > 100/min | 38 | 55 | 1.71(1.01-2.92) | 0.157 | 1.62(0.73-3.60) | 0.23 |
| Peripheral edema | No | 3 | 13 | 1 | | 1 | |
| | Yes | 83 | 161 | 2.23(0.62-8.06) | 0.182 | 0.64(0.1-4.38) | 0.65 |
| COPD | No | 77 | 167 | 1 | | 1 | |
| | Yes | 9 | 7 | 2.79(1.01-7.76) | 0.031 | 4.94(1.36-17.9) | **0.001*** |
| Anemia | No | 44 | 127 | 1 | | 1 | |
| | Yes | 42 | 47 | 2.58(1.5-4.42) | 0.189 | 1.18(0.55-2.53) | 0.67 |
| Hyponatremia | No | 20 | 150 | 1 | | 1 | |
| | Yes | 66 | 24 | 20.6(10.6-39.9) | 0.112 | 2.2(1.8-9.5) | **0.001*** |
| Hypokalemia | No | 56 | 161 | 1 | | 1 | |
| | Yes | 30 | 13 | 6.63(3.2-13.6) | 0.051 | 4.08(1.57-10.6) | **0.004*** |

*= indicate significance at <0.05, COPD: chronic obstructive pulmonary disease, NHYA: New York heart association, HF: Heart failure, HFpEF: Heart failure with preserved ejection fraction, HFrEF: Heart failure with reduced ejection fraction.

States (13.0%) [22], China (26.1%), and North America (18.0%). This disparity may be attributed to multiple factors. The extensive use of loop diuretics in our study population could be a major contributor, as these agents are known to increase urinary chloride excretion and significantly raise the risk of hypochloremia in heart failure patients [911]. Dietary differences may also play a role; for instance, in Japan, high chloride intake from seafood such as fish, kelp, and seaweed may protect against hypochloremia [23]. In contrast, limited dietary diversity and food insecurity in our setting may result in lower chloride intake. Other contributing factors may include delayed presentation of patients, more advanced disease severity on admission, differences in healthcare access, and variations in laboratory reference ranges and diagnostic criteria, all of which can influence the observed prevalence. Our finding aligns more closely with studies conducted in Ethiopia (36.7%), as well as others from the USA (31.5%) [10] and (36.0%) [24]. These similarities may reflect regional practices in diuretic use, nutritional status, and underlying patient characteristics [13,25].

Importantly, the study found that the median length of hospital stay was longer among patients with hypochloremia (12 days) compared to those with normal chloride levels (8.5 days), a difference that was statistically significant (p = 0.001). This observation aligns with previous research showing that hypochloremia is independently associated with prolonged

hospitalization in patients with acute and chronic heart failure [26]. The pathophysiological mechanisms underlying this association may include more severe neurohormonal activation, impaired renal function, and poor diuretic responsiveness among hypochloremic patients, all of which can complicate clinical management and delay recovery [27,9]. Hypochloremia may also serve as a surrogate marker for disease severity and ongoing volume overload, prompting clinicians to pursue more aggressive or prolonged treatment strategies. Moreover, the need for closer monitoring and frequent adjustments in diuretic therapy in patients with electrolyte imbalances may contribute to extended hospitalization. These findings highlight the association between hypochloremia and clinical outcomes in patients admitted with AHF. We suggest that further studies to explore whether early identification and management of electrolyte disturbances such as hypochloremia could independently influence patient outcomes and healthcare utilization. [28,29].

Moreover, in-hospital mortality was significantly higher among hypochloremic AHF patients (15.1%) compared to those with normal chloride levels (4.5%), with this difference reaching statistical significance (p = 0.003). This finding is consistent with studies from China, which reported a significantly increased 30-day mortality risk in hypochloremic AHF patients relative to those with normal chloride levels [28], as well as similar results observed in North America [30]. A comparable study conducted in Ethiopia also demonstrated higher in-hospital mortality among hypochloremic AHF patients [31]. The association between hypochloremia and poor outcomes may be explained by its effects on fluid balance and neurohormonal activation, which can exacerbate heart failure severity and contribute to diuretic resistance [12,9]. These mechanisms underscore the relevance of monitoring hypochloremia in patients with AHF. Collectively, our findings highlight the association between hypochloremia and adverse outcomes in AHF, suggesting areas for further research to evaluate potential clinical implications.

Furthermore, this study identified several factors significantly associated with hypochloremia: older age, NYHA class IV, history of COPD, hyponatremia, and hypokalemia. These findings are consistent with multiple studies. For example, older age has been consistently associated with an increased risk of hypochloremia in HF patients, possibly due to age-related renal function decline and altered electrolyte handling [32]. A study conducted in China involving 2,008 HF patients found that hypochloremia was significantly associated with more severe functional status (higher NYHA class), reflecting the greater neurohormonal activation and diuretic use in advanced HF that can lead to electrolyte imbalances [2226]. Comorbidities such as COPD and respiratory failure are more prevalent in hypochloremic patients, potentially might be because of chronic hypoxia and increased use of diuretics or corticosteroids that affect electrolyte balance [33,29]. Moreover, electrolyte disturbances, particularly hyponatremia and hypokalemia, commonly co-occur with hypochloremia in HF patients might indicate shared underlying mechanisms such as diuretic therapy and neurohormonal dysregulation [34,26].

While our study focused on hypochloremia at admission, we were unable to differentiate patients whose hypochloremia persisted during hospitalization from those whose levels normalized, which may have implications for outcomes. Future research could investigate this distinction and weather this analysed factors independent association with hypochloremia. Additionally, hypothesis-driven case-control studies with robust analytical models could further identify patient-level risk factors for hypochloremia, while interventional studies or randomized controlled trials could assess whether active correction influences clinical outcomes. Such investigations would help clarify whether hypochloremia is primarily a marker of disease severity or a modifiable factor in acute heart failure.

## Conclusion

In this cohort of adult patients with acute heart failure, the prevalence of hypochloremia at admission was 33.1%, with a mean serum chloride of 99.3 ± 8.4 mmol/L. In-hospital mortality was significantly higher among hypochloremic patients compared to normochloremic patients, and the median length of hospital stay was longer in hypochloremic patients (12 vs 8.5 days, p = 0.001). NYHA class IV, COPD, hyponatremia, and hypokalemia found associated with hypochloremia. Clinically, almost all patients received loop diuretics, often at substantial doses, and a high proportion presented with severe heart failure (NYHA III–IV). These findings suggest that hypochloremia may serve as a marker of disease severity and

higher risk of adverse outcomes, highlighting the need for careful electrolyte monitoring and tailored management, particularly in settings with high diuretic exposure and advanced HF.

## Limitations

This study limitation includes: its retrospective single-center design, reliance on chart documentation, potential selection bias due to inclusion only of patients with serum chloride measured, lack of standardized diuretic protocols and variable dosing, inability to assess post-discharge or long-term outcomes, and the absence of multivariable outcome modeling for mortality and length of stay.

## Supporting information

**S1 File. Hypochloremia - SPSS.**
(SAV)

## Author contributions

**Conceptualization:** Molla Asnake Kebede, Belay Mengistu Gebrie, Kedir Negesso Tukeni, Mohammed Mecha Abafogi, Turi Abateka Abadiga, Bethelhem Yaynemsa Sequr, Biniyam Beyene Tabor, Selemon Gebrezgabiher Asgedom.

**Data curation:** Molla Asnake Kebede, Belay Mengistu Gebrie, Kedir Negesso Tukeni, Mohammed Mecha Abafogi, Turi Abateka Abadiga, Bethelhem Yaynemsa Sequr, Biniyam Beyene Tabor, Selemon Gebrezgabiher Asgedom.

**Formal analysis:** Molla Asnake Kebede, Belay Mengistu Gebrie, Kedir Negesso Tukeni, Mohammed Mecha Abafogi, Turi Abateka Abadiga, Bethelhem Yaynemsa Sequr, Biniyam Beyene Tabor, Selemon Gebrezgabiher Asgedom.

**Funding acquisition:** Belay Mengistu Gebrie, Kedir Negesso Tukeni, Mohammed Mecha Abafogi, Bethelhem Yaynemsa Sequr.

**Investigation:** Belay Mengistu Gebrie, Kedir Negesso Tukeni, Mohammed Mecha Abafogi.

**Methodology:** Belay Mengistu Gebrie, Kedir Negesso Tukeni, Mohammed Mecha Abafogi, Turi Abateka Abadiga, Bethelhem Yaynemsa Sequr, Biniyam Beyene Tabor, Selemon Gebrezgabiher Asgedom.

**Project administration:** Belay Mengistu Gebrie, Kedir Negesso Tukeni, Mohammed Mecha Abafogi.

**Resources:** Molla Asnake Kebede, Belay Mengistu Gebrie, Kedir Negesso Tukeni, Mohammed Mecha Abafogi, Turi Abateka Abadiga, Bethelhem Yaynemsa Sequr, Biniyam Beyene Tabor.

**Software:** Molla Asnake Kebede, Belay Mengistu Gebrie, Kedir Negesso Tukeni, Mohammed Mecha Abafogi, Turi Abateka Abadiga, Bethelhem Yaynemsa Sequr, Biniyam Beyene Tabor, Selemon Gebrezgabiher Asgedom.

**Supervision:** Molla Asnake Kebede, Belay Mengistu Gebrie, Kedir Negesso Tukeni, Mohammed Mecha Abafogi, Turi Abateka Abadiga, Bethelhem Yaynemsa Sequr, Biniyam Beyene Tabor, Selemon Gebrezgabiher Asgedom.

**Validation:** Molla Asnake Kebede, Belay Mengistu Gebrie, Kedir Negesso Tukeni, Mohammed Mecha Abafogi, Turi Abateka Abadiga, Bethelhem Yaynemsa Sequr, Biniyam Beyene Tabor, Selemon Gebrezgabiher Asgedom.

**Visualization:** Molla Asnake Kebede, Belay Mengistu Gebrie, Kedir Negesso Tukeni, Mohammed Mecha Abafogi, Turi Abateka Abadiga, Bethelhem Yaynemsa Sequr, Biniyam Beyene Tabor, Selemon Gebrezgabiher Asgedom.

**Writing – original draft:** Molla Asnake Kebede, Belay Mengistu Gebrie, Kedir Negesso Tukeni, Mohammed Mecha Abafogi, Turi Abateka Abadiga, Bethelhem Yaynemsa Sequr, Biniyam Beyene Tabor, Selemon Gebrezgabiher Asgedom.

**Writing – review & editing:** Molla Asnake Kebede, Belay Mengistu Gebrie, Kedir Negesso Tukeni, Mohammed Mecha Abafogi, Turi Abateka Abadiga, Bethelhem Yaynemsa Sequr, Biniyam Beyene Tabor, Selemon Gebrezgabiher Asgedom.

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
