## [Decision Letter · Decision Letter 0]

11 Jan 2026

Dear Dr.  Kebede,

Thank you for submitting your manuscript to PLOS ONE. After careful consideration, we feel that it has merit but does not fully meet PLOS ONE’s publication criteria as it currently stands. Therefore, we invite you to submit a revised version of the manuscript that addresses the points raised during the review process.

The title of the manuscript is quite pertinent and merits attention. Although it is well-written, there are certain issues that should be addressed. The majority of these have been noted by the reviewers. However a few additional issues also need to be addressed.

Sometimes up to five references were cited to support an assertion. Ideally, one to three references is enough.  Five references suggest that you're unsure. I would advise not citing more than three references all at once.Over half of the studies' cited were written more than five years ago. At least 70% of the mentioned papers should ideally be under five years old. More current articles should be cited. Furthermore, some of the references do not follow the journal reference method.The ethical approval and consent to participate should be part of the Methods. There is also a need to delete the abbreviations and data availability from the manuscript.

We look forward to receiving your revised manuscript.

Kind regards,

Innocent Ijezie Chukwuonye, MBBS, FMCP(Internal Medicine)

Academic Editor

PLOS One

Journal Requirements:

“This research was funded by Jimma University solely to support the conduct of the study. No specific grants were received for authorship or publication from any public, commercial, or not-for-profit funding agencies.,”

4. Please note that funding information should not appear in any section or other areas of your manuscript. We will only publish funding information present in the Funding Statement section of the online submission form. Please remove any funding-related text from the manuscript.

5. We are unable to open your Supporting Information file “Hypochloremia - SPSS.sav”. Please kindly revise as necessary and re-upload.

7. Please upload a copy of Figure 3, to which you refer in your text on page 22. If the figure is no longer to be included as part of the submission please remove all reference to it within the text.

8. Please include captions for your Supporting Information files at the end of your manuscript, and update any in-text citations to match accordingly. Please see our Supporting Information guidelines for more information: http://journals.plos.org/plosone/s/supporting-information .

Reviewer's Responses to Questions

**Comments to the Author**

1. Is the manuscript technically sound, and do the data support the conclusions?

Reviewer #1: Partly

Reviewer #2: Partly

2. Has the statistical analysis been performed appropriately and rigorously?

Reviewer #1: No

Reviewer #2: I Don't Know

3. Have the authors made all data underlying the findings in their manuscript fully available?

Reviewer #1: Yes

Reviewer #2: No

4. Is the manuscript presented in an intelligible fashion and written in standard English?

Reviewer #1: Yes

Reviewer #2: No

Reviewer #1: The study addresses an important and underexplored question in acute heart failure (AHF) within a low‑resource African setting and provides clinically relevant data on the prevalence, correlates, and short‑term impact of hypochloremia. The main findings are plausible and broadly consistent with existing literature, but there are significant issues in design description, statistical analysis, clarity, and language that require major revision before the work is suitable for publication.

1. Major issues

1.1 Study design and internal consistency

The manuscript describes the design as an “institution-based cross-sectional study,” yet the analyses clearly involve time‑dependent in‑hospital outcomes (mortality and length of stay). In essence, the study is a retrospective observational cohort (retrospective chart review) of AHF admissions with follow‑up until discharge or death. This inconsistency can confuse readers and undercuts the methodological rigor.

Please re‑classify and consistently describe the design as a retrospective observational or retrospective cohort study of hospitalized AHF patients with serum chloride measured at admission.

Clarify the study flow: total number of AHF admissions during the period, number excluded for missing chloride or incomplete records, and final number analyzed. A simple flow diagram would be helpful.

1.2 Operational definitions and exposure–outcome structure

The manuscript provides detailed operational definitions (AHF, hypochloremia, hyponatremia, HF phenotypes, etc.), which is commendable. However, the roles of some defined constructs are not clearly integrated into the analysis:

“Corrected” vs “persistent” hypochloremia are defined but not meaningfully analyzed beyond basic counts.

The primary exposure appears to be hypochloremia at admission, but this is not explicitly declared as such in the Methods.

Please:

Explicitly state the primary exposure (hypochloremia at admission) and primary outcomes (in‑hospital mortality and length of hospital stay).

Either remove “corrected/persistent hypochloremia” from the operational definitions or include a clearly planned secondary analysis comparing their prognostic impact, if the data allow.

1.3 Statistical analysis and claims of prognostic independence

The descriptive analyses, Chi‑square test for mortality, and Mann–Whitney U test for length of stay are appropriate for unadjusted comparisons. The multivariable logistic regression for predictors of hypochloremia is also appropriate in principle, and the reported associations with NYHA class IV, COPD, hyponatremia, and hypokalemia are clinically plausible. However:

There is no multivariable model assessing whether hypochloremia independently predicts in‑hospital mortality or length of stay after adjustment for other markers of disease severity (age, NYHA class, ejection fraction, renal function, sodium, potassium, diuretic dose, comorbidities).

Despite this, parts of the Discussion and Conclusion imply that hypochloremia is an independent prognostic factor in this cohort.

To strengthen the manuscript and support prognostic claims, the following are strongly recommended:

Construct a multivariable logistic regression model for in‑hospital mortality including at minimum: age, sex, NYHA class, HF phenotype (HFrEF/HFpEF/HFmrEF), COPD, renal function (creatinine or eGFR), sodium, potassium, loop diuretic use/dose, and hypochloremia at admission.

Consider a multivariable model for length of stay (e.g., negative binomial/Poisson or linear regression on log‑transformed LOS) with the same covariates, including hypochloremia.

Clearly describe variable selection (clinical vs p‑value–based), check for multicollinearity, and report model fit indices (in addition to the Hosmer‑Lemeshow test).

If you are unable to perform adjusted outcome models, the wording throughout the manuscript should be revised to emphasize associations rather than independent prognostic effects, and this limitation should be explicitly acknowledged.

1.4 Sampling, sample size, and potential selection bias

The sample size calculation is described but not completely transparent. The text mentions an assumed prevalence of 36.7% and initial sample size of 357, then a finite population correction based on 700 AHF patients to reach 260, but the intermediate steps are not shown. Moreover, “systematic random sampling” is stated without sufficient detail.

Please show the exact formula and parameters used (P, Z, d, N) and the calculation steps leading to 357 and then 260.

Justify the estimate of 700 AHF patients (e.g., internal hospital statistics for the defined period).

Describe the systematic sampling method more clearly: how were records ordered, what was the sampling interval (k), how was the random starting point chosen, and how were ineligible or missing charts handled.

Discuss the potential for selection bias due to inclusion only of patients with serum chloride measured at admission and how this might affect generalizability.

2. Presentation of results

2.1 Descriptive characteristics and tables

The manuscript provides a good overview of baseline clinical, laboratory, and treatment characteristics. There are, however, several areas that require clarification or correction:

Typographical and wording errors (e.g., “A of total 260…”, “peripheral enema” instead of “peripheral edema”, “de nevo” instead of “de novo”) should be corrected throughout.

Table 1 and subsequent tables need to strictly follow journal style: include clear titles, units, denominators, and footnotes explaining any abbreviations.

For interpretability, a dedicated table comparing key baseline characteristics between hypochloremic and non‑hypochloremic groups (demographics, comorbidities, HF type, NYHA class, renal function, sodium, potassium, diuretic use and dose) would be very useful.

2.2 Main findings

The key results are clearly presented:

Prevalence of hypochloremia 33.1% (86/260), mean chloride 99.3 ± 8.4 mmol/L (range 70–136).

In‑hospital mortality significantly higher in hypochloremic patients (15.1%) compared to normochloremic patients (4.5%), p = 0.003.

Median length of stay longer in hypochloremic patients (12 days vs 8.5 days), p = 0.001.

In multivariable analysis, NYHA class IV, COPD, hyponatremia, and hypokalemia remain significantly associated with hypochloremia.

These findings are important and potentially practice‑influencing for settings similar to the study site, especially given the high loop diuretic exposure documented. The manuscript would benefit from explicitly highlighting the clinical context, such as treatment patterns (almost all on loop diuretics, substantial doses) and the severity of HF at presentation (high proportion of NYHA III–IV).

3. Discussion and interpretation

The Discussion appropriately situates the results in the context of prior work from Ethiopia, Asia, Europe, and North America and offers plausible pathophysiologic explanations (RAAS activation, diuretic resistance, neurohormonal activation, volume overload). Nonetheless, several aspects need refinement:

At times, the narrative appears to imply a causal relationship between hypochloremia and poor outcomes, whereas the current analysis is largely observational and unadjusted for key confounders.

Statements that hypochloremia is a “strong prognostic indicator” should be tempered or supported by adjusted outcome models as suggested above.

The concept of “non-medication related associated factors” in the title and abstract is not clearly developed in the Discussion; medications (loop diuretics, RAAS blockers, SGLT2 inhibitors) are described but not systematically analyzed as predictors of hypochloremia.

Recommended revisions:

Reframe conclusions to emphasize that hypochloremia is associated with worse in‑hospital outcomes and more severe clinical profiles, but that residual confounding cannot be excluded.

If medication effects are of special interest, include them explicitly in the multivariable model for hypochloremia and discuss their role relative to “non‑medication” factors; otherwise, consider removing “Non‑Medication Related” from the title.

Add a more explicit hypothesis‑generating statement that future prospective and interventional studies are needed to determine whether active correction of hypochloremia improves outcomes.

4. Limitations and generalizability

The limitations section is relatively brief and should be expanded. At minimum, please address:

Retrospective single‑center design and reliance on chart documentation, which may introduce misclassification of AHF diagnosis, comorbidities, and outcomes.

Potential selection bias due to inclusion only of patients with serum chloride measured, which may oversample more severe or more closely monitored cases.

Lack of standardized diuretic protocols and possible confounding by varying diuretic dose and duration.

Inability to assess post‑discharge or long‑term outcomes; the prognostic assessment is limited to in‑hospital events.

Absence (in current form) of multivariable outcome modeling to fully establish independent association between hypochloremia and mortality/length of stay.

These additions will improve transparency and help readers interpret the findings in the appropriate context.

5. Ethics, data availability, and reporting

Ethics approval and confidentiality measures are reported, which is essential. However, the ethics paragraph appears to contain an incomplete phrase (“in accordance with the Declaration of…”) and should be corrected to “Declaration of Helsinki” or similar, as applicable.

The Data Availability statement in the manuscript and the Editorial Manager fields both state that all relevant data are within the manuscript and supporting files, but the wording appears somewhat duplicated and fragmented. For a PLOS ONE submission, please:

Ensure a single, clear Data Availability statement is included in the manuscript, fully aligned with journal policy and exactly matching what is declared in the submission system.

If possible, consider depositing a de‑identified dataset in a suitable public repository and provide the link/DOI, which would increase transparency and reuse potential.

6. Language, style, and formatting

The manuscript contains numerous grammatical errors, awkward phrases, and typographical issues that impede readability. Examples include incorrect word choices, spacing, and inconsistencies in terms and abbreviations.

A thorough language and copy‑editing pass, preferably by a fluent English speaker or professional editor, is strongly recommended before resubmission.

Ensure consistent use of terminology and abbreviations (e.g., NYHA, HFpEF, HFrEF, HFmrEF, COPD, etc.), and define each abbreviation at first use.

Standardize reference formatting according to PLOS ONE guidelines; verify journal names, year, volume/issue, pages, and DOIs where applicable.

7. Overall evaluation

In summary, this manuscript addresses a relevant and insufficiently studied topic—the prevalence and short‑term impact of hypochloremia in AHF patients in a resource‑limited African setting—and contributes potentially important data. However, several methodological and reporting issues (design labeling, incomplete description of sampling and analysis, absence of adjusted models for outcomes, and language/formatting problems) need substantial revision. With careful attention to the points above, the study could provide a valuable addition to the literature on electrolyte disturbances and heart failure, particularly in low‑resource contexts.

Reviewer #2: Reviewer’s feedback based on research articles using the following criteria:

1. The study presents the results of original research.

a. The study met this criterion by collecting original data locally at the Jimma Medical Center (JMC) by including only patients on admission using serum chlorine level.

2. Results reported have not been published elsewhere.

a. Not sure it has been published elsewhere.

3. Experiments, statistics, and other analyses are performed to a high technical standard and are described in sufficient detail.

a. The manuscript reports a hospital-based cross-sectional study using retrospective medical records. The authors clearly describe the sampling approach, and the sample size appears adequate for the stated analyses. Data were extracted retrospectively using a structured checklist by trained residents (via Kobo Toolbox) under PI supervision. Quality assurance procedures are also described, including a 5% pretest at a separate hospital to improve clarity, routine/daily review, cross-checking, and PI oversight. However, the Methods do not explain how missing data were assessed or handled (e.g., extent of missingness, exclusions, or imputation), which limits reproducibility and may affect interpretation of the results.

4. Conclusions are presented in an appropriate fashion and are supported by the data.

a. For (i) and (ii), since this study is a cross-sectional study, authors should not overstate that correcting hypochloremia improves health outcome since authors did not directly test for in the study. My suggestion: they should be revised to show association only; authors should consider framing any hypotheses for future studies.

i. “Early recognition and correction of hypochloremia may improve outcomes in this population.”

ii. “These findings highlight the importance of early identification and correction 333 of electrolyte imbalances, particularly hypochloremia, to potentially improve outcomes in 334 patients with acute heart failure.

b. In the Discussion, the authors imply that hypochloremia affects healthcare costs and system burden; however, cost/resource utilization outcomes were not measured in this study. Please revise this statement to reflect with the data in the study (i.e., the observed difference in length of stay) and frame any cost/burden implications as speculative or remove them.

5. The article is presented in an intelligible fashion and is written in standard English.

Article is understandable and readable, but I think authors need to pay more attention to the ethical statement part and result section as they might appear to be some grammatically error, and punctuation issue. “Participants names were not be recorded…” and “The study was conducted in accordance with the Declaration of…” that part seem incomplete.

6. The research meets all applicable standards for the ethics of experimentation and research integrity.

Manuscript met all ethical declaration the ethics statement should be revised for clarity (one sentence is incomplete), and the authors should consider adding the IRB approval reference/number and approval date to improve transparency.

7. The article adheres to appropriate reporting guidelines and community standards for data availability.

Authors should clarify what can be shared (de-identified dataset, codebook, analysis code) and the process/approvals required considering their data is clinical.

**Do you want your identity to be public for this peer review?** For information about this choice, including consent withdrawal, please see our Privacy Policy

Reviewer #1: **Yes:** Deepanshu

Reviewer #2: No

---

## [Author Response · Author response to Decision Letter 1]

24 Jan 2026

Editor comment

1- Thank you for submitting your manuscript to PLOS ONE. After careful consideration, we feel that it has merit but does not fully meet PLOS ONE’s publication criteria as it currently stands. Therefore, we invite you to submit a revised version of the manuscript that addresses the points raised during the review process.

Answer

We thank the Editor and reviewers for their careful evaluation of our manuscript and for recognizing its merit. We have revised the manuscript thoroughly to address all points raised during the review process. We believe that these revisions have substantially improved the clarity, rigor, and overall quality of the manuscript, and we hope that the revised version now meets the publication criteria of PLOS ONE.

2- The title of the manuscript is quite pertinent and merits attention. Although it is well-written, there are certain issues that should be addressed. The majority of these have been noted by the reviewers. However a few additional issues also need to be addressed.

Answer:

We thank the Editor for the positive assessment of the manuscript title and overall writing quality. We have carefully addressed all issues raised by the reviewers, as well as the additional concerns noted by the Editor. Appropriate revisions have been made throughout the manuscript to improve clarity, consistency, and scientific rigor. We believe these changes have strengthened the manuscript substantially.

3- Sometimes up to five references were cited to support an assertion. Ideally, one to three references are enough. Five references suggest that you're unsure. I would advise not citing more than three references all at once.

Answer: We thank the reviewer for this helpful suggestion. We have revised the manuscript to limit the number of references supporting a single assertion to a maximum of three, citing only the most relevant and authoritative sources.

4- Over half of the studies cited were written more than five years ago. At least 70% of the mentioned papers should ideally be under five years old. More current articles should be cited. Furthermore, some of the references do not follow the journal reference method.

Answer;

Thank you for your comment. We have revised the references in the manuscript to ensure that the majority are from the last five years, with at least 70% of cited studies being recent and directly relevant. We have also carefully reformatted all references to fully comply with the PLOS ONE referencing style, including journal names, volume, issue, pages, and DOI numbers where available.

5- The ethical approval and consent to participate should be part of the Methods. There is also a need to delete the abbreviations and data availability from the manuscript.

Answer

We thank the Editor for this important comment. The information on ethical approval and consent to participate has now been moved to the Methods section. In addition, the Abbreviations and Data Availability sections have been removed from the manuscript in accordance with the journal’s requirements.

6- A letter that responds to each point raised by the academic editor and reviewer(s). You should upload this letter as a separate file labeled 'Response to Reviewers'.

Answer

We thank the Editor for this instruction. A detailed point-by-point response addressing all comments raised by the Academic Editor and the reviewers has been prepared and uploaded as a separate file entitled “Response to Reviewers.”

7- A marked-up copy of your manuscript that highlights changes made to the original version. You should upload this as a separate file labeled 'Revised Manuscript with Track Changes'.

Answer: We thank the Editor for this instruction. A marked-up version of the manuscript highlighting all revisions has been prepared and uploaded as a separate file entitled “Revised Manuscript with Track Changes.”

8- An unmarked version of your revised paper without tracked changes. You should upload this as a separate file labeled 'Manuscript'.

Anwer

We thank the Editor for this instruction. An unmarked version of the revised manuscript, without tracked changes, has been prepared and uploaded as a separate file entitled “Manuscript.”

9- 10- If applicable, we recommend that you deposit your laboratory protocols in protocols.io to enhance the reproducibility of your results. Protocols.io assigns your protocol its own identifier (DOI) so that it can be cited independently in the future. For instructions see: https://journals.plos.org/plosone/s/submission-guidelines#loc-laboratory-protocols. Additionally, PLOS ONE offers an option for publishing peer-reviewed Lab Protocol articles, which describe protocols hosted on protocols.io. Read more information on sharing protocols at https://plos.org/protocols?utm_medium=editorial-email&utm_source=authorletters&utm_campaign=protocols.

Answer:

We appreciate the suggestion. While our study did not involve novel laboratory protocols, all procedures followed standard clinical and laboratory practices. We have ensured that methods are described in sufficient detail to allow reproducibility. Since no unique protocols were developed, deposition in protocols.io was not applicable.

11- Please ensure that your manuscript meets PLOS ONE's style requirements, including those for file naming. The PLOS ONE style templates can be found at

Answer

We have ensured that the manuscript fully complies with PLOS ONE style requirements, including formatting and file naming conventions. The manuscript has been prepared using the official PLOS ONE style templates, which are available on the PLOS ONE website.

12- We note that the grant information you provided in the ‘Funding Information’ and ‘Financial Disclosure’ sections do not match. When you resubmit, please ensure that you provide the correct grant numbers for the awards you received for your study in the ‘Funding Information’ section.

Answer:

We appreciate your observation. We have reviewed the sections and corrected the discrepancies. The Funding Information and Financial Disclosure sections now consistently reflect the same grant source, number, and supporting institution.

13- .Thank you for stating the following financial disclosure: “This research was funded by Jimma University solely to support the conduct of the study. No specific grants were received for authorship or publication from any public, commercial, or not-for-profit funding agencies.,”

Answer

Funding

This research was funded by Jimma University solely to support the conduct of the study. No specific grants were received for authorship or publication from any public, commercial, or not-for-profit funding agencies. The funders had no role in study design, data collection and analysis, decision to publish, or preparation of the manuscript Reference: JMU 222/25

Accept and corrected

14- Please state what role the funders took in the study. If the funders had no role, please state: "The funders had no role in study design, data collection and analysis, decision to publish, or preparation of the manuscript." If this statement is not correct you must amend it as needed.

Answer

Accept and corrected as provides

15- Please include this amended Role of Funder statement in your cover letter; we will change the online submission form on your behalf.

Answer

Accept and corrected as provides

16- Please note that funding information should not appear in any section or other areas of your manuscript. We will only publish funding information present in the Funding Statement section of the online submission form. Please remove any funding-related text from the manuscript.

Answer

Accept and corrected as provides

15. We are unable to open your Supporting Information file “Hypochloremia - SPSS.sav”. Please kindly revise as necessary and re-upload.

Answer

Accept and corrected as provides

16. Your ethics statement should only appear in the Methods section of your manuscript. If your ethics statement is written in any section besides the Methods, please move it to the Methods section and delete it from any other section. Please ensure that your ethics statement is included in your manuscript, as the ethics statement entered into the online submission form will not be published alongside your manuscript.

Answer

Accept and corrected as provides

17. Please upload a copy of Figure 3, to which you refer in your text on page 22. If the figure is no longer to be included as part of the submission please remove all reference to it within the text.

Answer

Accept and corrected as provides

18. Please include captions for your Supporting Information files at the end of your manuscript, and update any in-text citations to match accordingly. Please see our Supporting Information guidelines for more information: http://journals.plos.org/plosone/s/supporting-information.

Answer: We have added captions for all Supporting Information files at the end of the manuscript and updated the corresponding in-text citations to match, in accordance with PLOS ONE’s Supporting Information guidelines.

Answer

We thank the reviewer for the suggested references. We carefully reviewed the recommended publications and cited those that are directly relevant and contribute to the context and interpretation of our findings. References that were not cited were determined to be outside the scope of the present study.

Review Comments to the Author

Reviewer #1: The study addresses an important and underexplored question in acute heart failure (AHF) within a low‑resource African setting and provides clinically relevant data on the prevalence, correlates, and short‑term impact of hypochloremia. The main findings are plausible and broadly consistent with existing literature, but there are significant issues in design description, statistical analysis, clarity, and language that require major revision before the work is suitable for publication.

Answer;

We thank the reviewer for acknowledging the clinical relevance of our study. We have addressed the concerns by revising the study design description, clarifying the statistical analysis, and improving language and overall manuscript clarity. These changes enhance the rigor and readability of the work.

1. Major issues

1.1 Study design and internal consistency

The manuscript describes the design as an “institution-based cross-sectional study,” yet the analyses clearly involve time‑dependent in‑hospital outcomes (mortality and length of stay). In essence, the study is a retrospective observational cohort (retrospective chart review) of AHF admissions with follow‑up until discharge or death. This inconsistency can confuse readers and undercut the methodological rigor.

Answer

Thank you for the comment. We have clarified the study design and now consistently describe the study as a retrospective observational cohort study throughout the manuscript, including the Methods section.

Please re‑classify and consistently describe the design as a retrospective observational or retrospective cohort study of hospitalized AHF patients with serum chloride measured at admission.

Answer

Thank you for this comment. We have reclassified and consistently described the study design as a retrospective observational cohort study of hospitalized patients with acute heart failure, with serum chloride measured at admission, throughout the manuscript (including the Abstract and Methods sections).

Clarify the study flow: total number of AHF admissions during the period, number excluded for missing chloride or incomplete records, and final number analyzed. A simple flow diagram would be helpful.

Answer:

Thank you for this helpful suggestion. We have clarified the study flow by reporting the total number of acute heart failure (AHF) admissions during the study period, the number of patients excluded due to missing serum chloride measurements or incomplete records, and the final number included in the analysis. In addition, we have added a simple flow diagram to visually illustrate the patient selection process.

1.2 Operational definitions and exposure–outcome structure

The manuscript provides detailed operational definitions (AHF, hypochloremia, hyponatremia, HF phenotypes, etc.), which is commendable. However, the roles of some defined constructs are not clearly integrated into the analysis:

“Corrected” vs “persistent” hypochloremia are defined but not meaningfully analyzed beyond basic counts.

Answer

Thank you for this comment. We agree with the reviewer that corrected and persistent hypochloremia were not meaningfully analyzed in the present study. As the analysis was limited to hypochloremia at admission, as we also revised the Methods section accordingly and removed the definitions of corrected and persistent hypochloremia from the operational definitions to ensure consistency between the study objectives, methods, and analyses.

The primary exposure appears to be hypochloremia at admission, but this is not explicitly declared as such in the Methods.

Answer

Thank you for this comment. We have revised the Methods section to explicitly state that the primary exposure in this study is hypochloremia at admission. This clarification ensures that the study objectives, exposure definition, and analysis are clearly aligned.

Explicitly state the primary exposure (hypochloremia at admission) and primary outcomes (in‑hospital mortality and length of hospital stay).

Answer.

Thank you for this comment. We have revised the Methods section to explicitly state that the primary exposure is hypochloremia at admission, and the primary outcomes are in-hospital mortality and length of hospital stay. This clarification ensures alignment between the study objectives, exposure, and outcome measures.

Either remove “corrected/persistent hypochloremia” from the operational definitions or include a clearly planned secondary analysis comparing their prognostic impact, if the data allow.

Answer

Thank you for this suggestion. As the present study focuses on hypochloremia at admission and we did not perform a secondary analysis of corrected or persistent hypochloremia, we have removed these terms from the operational definitions to maintain consistency between the study objectives, methods, and analysis.

1.3 Statistical analysis and claims of prognostic independence

The descriptive analyses, Chi‑square test for mortality, and Mann–Whitney U test for length of stay are appropriate for unadjusted comparisons. The multivariable logistic regression for predictors of hypochloremia is also appropriate in principle, and the reported associations with NYHA class IV, COPD, hyponatremia, and hypokalemia are clinically plausible. However: There is no multivariable model assessing whether hypochloremia independently predicts in‑hospital mortality or length of stay after adjustment for other markers of disease severity (age, NYHA class, ejection fraction, renal function, sodium, potassium, diuretic dose, comorbidities). Despite this, parts of the Discussion and Conclusion imply that hypochloremia is an independent prognostic factor in this cohort. To strengthen the manuscript and support prognostic claims, the following are strongly recommended: Construct a multivariable logistic regression model for in‑hospital mortality including at minimum: age, sex, NYHA class, HF phenotype (HFrEF/HFpEF/HFmrEF), COPD, renal function (creatinine or eGFR), sodium, potassium, loop diuretic use/dose, and hypochloremia at admission. Consider a multivariable model for length of stay (e.g., negative binomial/Poisson or linear regression on log‑transformed LOS) with the same covariates, including hypochloremia. Clearly describe variable selection (clinical vs p‑value–based), check for multi

---

## [Editor Report · Decision Letter 1]

24 Feb 2026

Hypochloremia, Non-Medication Related Associated Factors, and Impact on Clinical Outcomes in Patients with Acute Heart Failure: Insights from Resource limited setups

PONE-D-25-64426R1

Dear Dr. ,**Molla Asnake Kebede**

We’re pleased to inform you that your manuscript has been judged scientifically suitable for publication and will be formally accepted for publication once it meets all outstanding technical requirements.

Kind regards,

Innocent Ijezie Chukwuonye, MBBS, FMCP (Internal Medicine)

Academic Editor

PLOS One

---

## [Editor Report · Acceptance letter]

PONE-D-25-64426R1

PLOS One

Dear Dr. Kebede,

I'm pleased to inform you that your manuscript has been deemed suitable for publication in PLOS One. Congratulations! Your manuscript is now being handed over to our production team.

Kind regards,

on behalf of

Dr. Innocent Ijezie Chukwuonye

Academic Editor

PLOS One